# Subclinical Mastitis Detected during the Last Gestation Period Can Increase the Risk of Stillbirth in Dairy Calves

**DOI:** 10.3390/ani12111394

**Published:** 2022-05-28

**Authors:** Ramūnas Antanaitis, Vida Juozaitienė, Vesta Jonike, Walter Baumgartner, Algimantas Paulauskas

**Affiliations:** 1Large Animal Clinic, Veterinary Academy, Lithuanian University of Health Sciences, LT-47181 Kaunas, Lithuania; 2Department of Biology, Faculty of Natural Sciences, Vytautas Magnus University, 44248 Kaunas, Lithuania; vida.juozaitiene@vdu.lt (V.J.); vesta.jonike@vdu.lt (V.J.); algimantas.paulauskas@vdu.lt (A.P.); 3University Clinic for Ruminants, University of Veterinary Medicine, A-1210 Vienna, Austria; walter.baumgartner@vetmeduni.ac.at

**Keywords:** mastitis, stillbirth, last gestation period

## Abstract

**Simple Summary:**

The aim was to investigate the relation of subclinical mastitis detected during the last gestation period and its pathogens with stillborn calves, considering that parity and herd size may also affect this result. This study shows that the late gestation period is challenging for stillbirth in next lactation. Collectively, these results suggest that decreasing incidence of subclinical mastitis during the last gestation period (from the 210th day of pregnancy) can decrease the risk of stillbirth in dairy calves. Further, it is important to identify the pathogen because the highest risk of stillbirth was found in cows with mastitis caused by *Escherichia coli, Staphylococcus aureus, Streptococcus agalactiae*, pathogenic Staphylococci and other Streptococci. Cows at the first calving were 1.38–1.65-times higher risk for the stillbirth of calves than in cows of parity ≥ 2.

**Abstract:**

We hypothesized that subclinical mastitis detected during the last gestation period can increase the risk of stillbirth in dairy calves. The aim was to investigate the relation of subclinical mastitis detected during the last gestation period and its pathogens with the stillbirth of calves. Cows from the 210th day of pregnancy were selected for the study. They were divided into two groups: the first group—subclinical mastitis was confirmed on the farm by the California mastitis test (CMT); the second group of cows—mastitis was not confirmed by the CMT test. Groups of cows were compared according to the results of their calving—the number of stillborn calves. A stillborn calf was defined as a calf that dies at birth or within the first 24 h after calving, following a gestation period of 260 days. Our results suggest that decreasing the incidence of subclinical mastitis during the last gestation period (from the 210th day of pregnancy) can decrease the risk of stillbirth in dairy calves. Further, it is important to identify the pathogen because the highest risk of stillbirth was found in cows with mastitis caused by *Escherichia coli*, *Staphylococcus aureus*, *Streptococcus agalactiae*, pathogenic Staphylococci and other Streptococci. Cows at the first calving had a 1.38–1.65-times higher risk of having stillborn calves than cows of parity ≥ 2. From a practical point, veterinarians and farmers can consider the effect of subclinical mastitis during late gestation on the risk of stillbirth and it could help for strategies of optimizing reproductive performance in dairy cows.

## 1. Introduction

Dairy farmers all across the world are working to increase their efficiency by boosting production per cow and, as a result, lowering the cost per unit produced [1]. Parturition is a traumatic experience for both cows and their calves [2]. Stillbirth (SB) is described as calf death occurring immediately before, during, or after parturition [3]. When the negative impacts of stillbirth on the cows’ milking performance are included, the economic loss is significantly higher. Stillbirths were linked to an increased risk of developing metritis and a retained placenta, as well as a lower risk of ovulatory infertility or of the oocyte being fertilized [4]. Stillbirth considerably reduces daily milk production, and the effect (a loss of 1.1 kg/day) was comparable to other disorders with known consequences (such as mastitis and lameness). The negative effect of stillbirth on milk production was greatest on early lactation [5]. Because it limits the number of animals available for sale, calf mortality is an important economic component of the farming system [6]. Calving difficulty has been proven to be significantly connected with an increased risk of SB [7].

Mastitis is a complicated and costly disease of dairy cows that dramatically lowers milk output and dairy sector profitability [8].

However, clinical mammary tissue will nearly invariably contain SCC with more than 200,000 cells per milliliter. Milk from healthy, uninfected mammary glands is white to whitish-yellow in color and is devoid of flakes, clots, and other obtrusive modifications in appearance. The great majority of these defects are caused by mammary gland bacterial infection. In general, the aberrant appearance of the secretion from the infected area increases with the severity of the infection. When quarterly SCC is equivalent to or greater than 200,000 cells/mL and bacteria are identified in the absence of clinical changes, the quarter is considered subclinical [9].

The immune response to a mammary gland infection is of the highest relevance for the dairy cow’s health. Harmon [10] observed that the rise in milk SCC is due to the migration of polymorphonuclear cells from the blood vessels to the mammary gland, as a result of the production of inflammatory mediators. This mechanism may be comparable in animals with subclinical mastitis, resulting in diminished reproductive performance [11].

It is estimated that subclinical mastitis is more economically significant than symptomatic mastitis [10]. Subclinical mastitis is more difficult to identify and leads to increased output losses. Schrick et al. [11] reported that the detrimental effects of mastitis on the reproductive efficiency of dairy cows are not restricted to the clinical form of the disease but may also be seen when the disease is in its subclinical stage. Hawari and Dabbas [12] found that *Streptococcus agalactiae* and *Staphylococcus aureus* have been identified as prevalent infectious organisms related with mastitis in dairy cows. Gitau et al. [13] estimated that coliform bacteria and environmental Streptococci are also abundant in cows’ environments. Reducing perinatal mortality is critical for improving overall welfare on dairy farms and sustaining customer trust globally [14]. However, the temporal connection between subclinical mastitis and stillbirth was not thoroughly investigated because the risk period for mastitis (1 to 90 days of gestation) overlapped with the follow-up period (45 to 270 d after artificial insemination (AI)) [14]. More research is needed to compare the effect of mastitis on pregnancy loss (PL) in dairy cows, before and during gestation, as well as the influence of mastitis on PL in dairy cows during different lactations [14].

In our past studies, we found that stillbirth was 11.2-times greater in cows with significantly difficult calving compared to cows with minimal or minor calving difficulties. Dystocia impaired lactation performance and increased the somatic cell count and the occurrence of mastitis, especially mastitis caused by *Streptococcus agalactiae* and *Staphylococcus aureus* [15].

According to analysis in the literature, we did not find any available information about how cows with mastitis during the last gestation period may be unfavorably associated with stillborn calves. The aim was to investigate the relation of subclinical mastitis detected during the last gestation period and its pathogens with stillborn calves, considering that parity and herd size may also affect this result.

## 2. Materials and Methods

### 2.1. Location and Animals

The study (in accordance with the provisions of the Law on Animal Welfare and Protection of the Republic of Lithuania; the study approval number is PK016965) was conducted on 10 farms of the Association of Lithuanian Black-and-White Breeders in 2015–2021.

In selected herds, the average somatic cell count in milk over the past three months was above 200,000 cells/mL, and the average cow herd production exceeded 8000 kg of milk. The dry period of cows lasted from 45 to 60 days.

The cows in the herds were milked by Lely Astronaut^®^ A3 robots with free traffic. The cows were of Holstein breed and their feed ration on all farms was balanced to match the energy and nutrient requirements of a Lithuanian Black-and-White cow weighing 550–650 kg, producing an average of 30 kg of milk per day. All farms used a zero-grazing system.

The herds of cows were divided into two classes according to the average annual number of the herd (the first class—100–200 cows (the first–fifth herds), the second—201–600 cows (the sixth–tenth herds)).

Cows from the 210th day of pregnancy were selected for the study. They were divided into two groups: the first group—subclinical mastitis was confirmed on the farm by the California mastitis test (CMT); the second group of cows—mastitis was not confirmed by the CMT test.

For this study, we used the most commonly used SCC diagnostic, the California Mastitis Test and various CMT score cutoff points were utilized to determine a positive CMT reaction [16]. The single milk bacteriological culture result was used as the gold standard for calculating diagnostic test features [17]. CMT was performed on each udder quarter of all cows. The CMT results were classified as negative (0+) or positive (1+) (traces), 2+ (gel), and 3+. (clumps) [17]. Milk samples were collected aseptically from CMT >1+ quarters and submitted for somatic cell counting (SCC), bacteriological testing in milk (using Somascope, CA-3A4, Delta Instruments, The Netherlands) at the state enterprise Pieno Tyrimai. Milk samples with a negative test for CMT were evaluated in the Pieno Tyrimai laboratory for the number of somatic cells.

After such testing of milk samples, groups of cows were finally formed. Cows with isolated pathogens from the mammary glands were assigned to the group of subclinical mastitis (*n* = 3582). The number of somatic milk cells in their samples exceeded 200,000/mL. and the average value for the group was 581,000/mL. The average number of somatic cells in the milk of cows with a negative test result for CMT was 99,980/mL. They were defined as a group of healthy cows (*n* = 1593) whose data were used for comparison. Cows with a positive CMT result but no isolated pathogen were excluded from this study. Healthy cows in terms of reproductive status were similar to those with mastitis.

We evaluated cows in one herd during the study period: 322 to 701 with established mastitis and 234 to 520 healthy. Samples of sick and healthy animals in the herds were: 242 and 322 (Herd 1), 234 and 341 (Herd 2), 324 and 387 (Herd 3), 331 and 396 (Herd 4), 344 and 499 (Herd 5), 317 and 367 (Herd 6), 389 and 586 (Herd 7), 398 and 597 (Herd 8), 483 and 674 (Herd 9), 520 and 701 (Herd 10) cows, respectively (Table 1).

Healthy cows (*n* = 1593) were also compared with groups of animals with isolated mastitis pathogens: Enterococci (*n* = 87), *Escherichia coli* (*n* = 79), mixed microbiota (*n* = 565), non-pathogenic Staphylococci (*n* = 381), other Gram-negative species (*n* = 374), other Gram-positive species (*n* = 382), other Streptococci (*n* = 303), pathogenic Staphylococci (*n* = 399), Serogroup C Streptococci (*n* = 79), Serogroup D Streptococci (*n* = 82), Serogroup G Streptococci (*n* = 87), Staphylococcus aureus (*n* = 380), and Streptococcus agalactiae (*n* = 384). Cows with a positive CMT reaction were treated to all quarters with Rilexine DC (375 mg of cephaleksin, Virbac S.A.1ère Avenue, 2065 m, L.I.D. 06,516 Carros, France). Internal teat sealant (ITS) containing bismuth subnitrate was infused into all quarters of all cows (Orbeseal, Zoetis). Following the final milking, antibiotic and ITS infusions were given as follows: teat ends were cleansed for at least 5 s with 70% isopropyl alcohol-soaked cotton swabs by trained staff wearing clean disposable gloves before the antibiotic treatment was infused into the mammary gland and again before ITS was infused into the teat cistern [9].

Cows were also classified by parity: parity 1 (*n* = 2425; 722 healthy and 1703 cows with subclinical mastitis) and parity ≥ 2 (*n* = 2750; 871 healthy and 1879 cows with subclinical mastitis).

The herds of cows were divided into two classes according to the size of the herd: class 1—100–200 cows (average 126 cows in the herd) class 2—201–600 cows (average 351 cows in the herd).

Groups of cows were compared according to the results of their calving—the number of stillborn calves. A stillborn calf was defined as a calf that dies at birth or within the first 24 h after calving following a gestation period at least 260 days.

### 2.2. Measurements

To study microbial diversity four to five milk streams were collected in a tube containing a boric-acid-based preservative (Merck KGaA, Darmstadt, Germany). Each teat was washed with a tissue moistened in a 70% ethanol solution and the first two to three streams of milk were discharged before the sample was taken [18]. Milk samples were obtained from all quarters of the cow’s udder in the same tube. The samples were kept at a temperature of 4 ± 2 °C for no more than three days. The detection of subclinical mastitis-related bacteria in milk samples from cows was carried out at the state enterprise “Pieno Tyrimai” according to the method recommended by Kroger and Jasper [18]

For the initial screening 10 µL milk samples were cultivated in a blood agar base with esculin and incubated for 24–72 h at temperature (37 ± 1 °C) to determine the type of haemolysis. The appearance, size, colour, and haemolysis zones of grown colonies were evaluated visually. Microorganisms were then classified using potassium hydroxide and Gram colouring into yeast, Gram-positive and Gram-negative bacteria, and cocci or bacilli. Catalase test was used to distinguish between Staphylococcus (Staph.) and Streptococcus (Strep.). Bacteria, which produce coagulase, ferment mannitol and grow on Baird-Parker agar in black colonies with a clear halo, were considered to be *Staph. aureus*. Hydrolysis of esculin and Lancefield grouping were used to identify Streptococcus. If esculin hydrolysis was detected, bacteria were processed with a pyrrolidonyl aminopeptidase (PYR) test. Enterococcus was identified when positive PYR test results were obtained. Drigalski and Chromocult Coliform agar tests were done for Gram-negative bacilli identification. Bacteria isolated from Drigalski and Chromocult Coliform agar, as well as former dark blue or purple colonies, were identified as *Escherichia coli*. Mixed microbiota refers to samples that contain more than one microorganism species [9,19].

### 2.3. Data Analysis and Statistics

Statistical analysis of this study data was performed using SPSS 25.0 software (IBM Corp. released 2017. IBM SPSS Statistics for Windows, Version 25.0. Armonk, NY, USA: IBM Corp.)

Pearson’s chi-square test was used to determine if there was a statistically significant difference between category frequencies in the groups.

The number of stillborn calves in healthy cows was compared with (a) the number of stillborn calves in cows with subclinical mastitis, (b) the number of stillborn calves in cows with known mastitis pathogens.

Multivariable binary logistic regression was used to predict the risk of stillborn calves in cows by parity (parity 1, parity ≥ 2) and healthy status (detected or not detected mastitis), herd size class (100–200 cows and 201–600 cows) and mastitis agent (grouped by mastitis agent or healthy cows). A total of 13 categories (groups) was used for isolated pathogens of mastitis from the mammary gland of cows. Multivariable logistic regression models were used to analyse the factors influencing the likelihood of stillbirth in calves, using a backward stepwise logistic regression method to eliminate all non-essential explanatory variables. Variables were constantly removed from the models according to the significance of the Wald criterion. All final statistical models included only significant explanatory variables.

A probability of less than 0.05 was considered significant (*p* < 0.05) for all tests used.

## 3. Results

We found that 8.9% of cows had subclinical mastitis during the last gestation period. The study found approximately 5.8% of stillborn calves in healthy cows (from 5.0 to 5.5% in class 1 herds and from 5.7 to 6.4% in class 2 herds) and 11.8% in the mastitis group (from 10.9 to 11.6% in class 1 herds and from 11.9 to 12.2% in class 2 herds) (*p* < 0.001).

The lowest number of calves born alive or surviving within 24 h of calving was found in cows with mastitis pathogens, such as *E. coli*, pathogenic Staphylococci, *Staphylococcus aureus*, *Streptococcus agalactiae* and serogroup G Streptococci, at the end of pregnancy. The percentage of stillborn calves in such cows ranged from 13.5 to 18.2 of the number of calves born (Figure 1).

As Figure 2 identifies, there is a wide spread in the number of stillborn calves by parity. The number of stillborn calves from healthy cows was 1.58-times lower in primiparous and 2.45-times lower in multiparous cows compared to the mastitis group by appropriate parity (*p* < 0.001). The analysis showed that the stillbirth of healthy primiparous cows was 1.63-times higher than that of multiparous cows (*p* = 0.05); moreover, the difference between the groups of cows with identified mastitis pathogens by parity was smaller at 1.06-times.

The greatest difference between cows was the identified mastitis agents by parity, with *Staphylococcus aureus, Escherichia coli*, serogroup C Streptococci pathogens, in which the stillbirth of calves at the first calving was 1.38–1.65-times higher (*p* < 0.05) than cows of parity ≥ 2 (Figure 3).

The binary logistic regression analysis showed that mastitis increased the risk of stillbirth in calves (1.467-fold, *p* < 0.001). The parity and herd class effects in the regression analysis models did not show a significant association with calf stillbirth. (Table 2).

When comparing a group of healthy cows with groups of cows for isolated mastitis pathogens from the mammary gland, it was found that most mastitis pathogens (except for enterococci, streptococci of serogroups C, D and G, non-pathogenic staphylococci) significantly increased the risk of stillborn calves (Table 3).

The analysis showed that the highest risk of stillbirth was found in cows with mastitis caused by *Escherichia coli*, *Staphylococcus aureus*, *Streptococcus agalactiae*, pathogenic Staphylococci and other Streptococci (2.469–2.905-times, *p* < 0.001).

An increase in parity significantly reduced the risk of stillbirth in cows with mastitis pathogens *Staphylococcus aureus* and *Streptococcus agalactiae* and non-pathogenic *Staphylococci* detected in the last period of gestation (0.630–0.716-times, *p* ≤ 0.05). The analysis showed that the herd class significantly increased the risk of stillbirth in cows with mastitis caused by Staphylococcus aureus (1.278-times, *p* = 0.05) and non-pathogenic Staphylococci (0.658-times, *p* = 0.05).

## 4. Discussion

During this study, we investigated the relation of subclinical mastitis detected during the last gestation period (from the 210th day of pregnancy) with the stillbirth of calves. According to our knowledge, most of the available research reports the impact of risk factors during the dry period on diseases after calving, such as subclinical mastitis, but there is limited information regarding how cows with mastitis during the last gestation period may be unfavourably associated with stillborn calves. In this study, we found that 8.9% of cows had subclinical mastitis during late pregnancy. 

According to recent research, the prevalence of bovine perinatal death is rising, notably among Holstein primiparae [20]. Bovine mortality is growing, particularly during the prenatal period, emphasizing the significance of benchmarking and identifying potential risk factors [21]. The most important modifiable variables increasing the risk of perinatal mortality are those that contribute to a higher prevalence of dystocia [20].

Mee et al. [20] found that perinatal mortality in dairy cattle is increasingly being recognized as a welfare issue. The most important modifiable variables impacting the risk of perinatal mortality are those that increase the likelihood of dystocia. By controlling predictors, such as breeding method, age at first calving, calving management, etc.), we can reduce the severity of this problem [20].

In our study, we found that approximately 5.8% of calves die in the first 24 h after birth in healthy cows and 11.8% in the mastitis group. According to the literature, the overall perinatal mortality rate ranged from 2.4 to 9.7% across the 26 trials (median 6.7%). Dhakal et al. described that each of the 26 studies classified perinatal death as stillbirth, though one included abortions as well [22]. Berry et al. [3] found that although the causes and concerns for stillbirths and live animal fatalities differ from those for abortions, 11 of the studies that did not include abortions did not describe how they excluded abortions or decided whether a stillbirth was not an abortion. Time periods utilized to define perinatal death in the 21 studies that indicated a time endpoint were 1 h (one study), 24 h (14 studies), and 48 h (six studies) and the relative mean mortality percentages for these study time periods were 3.5%, 6.4%, 6.6%, and 8% [22].

We found that the number of stillborn calves from healthy cows (without subclinical mastitis detected during the last gestation period) was 1.58-times lower in primiparous and 2.45-times lower in multiparous cows compared to the mastitis group by appropriate parity. According to the literature, the majority of diseases in dairy cows occur during or shortly after calving, which is a period of immunological suppression that results in an increased vulnerability to infections. The immunological response to a mammary gland infection is critical for the health of the dairy cow [10]. Schrick et al. [11] discovered that the rise in milk SCC is caused by the migration of polymorphonuclear cells from blood arteries to the mammary gland in response to inflammatory mediator release. This mechanism may be comparable in animals with subclinical mastitis, which causes reduced reproductive success [11].

Hernandez et al. [23] found that in dairy cows, subclinical mastitis is a risk factor for PL. The risks of PL were 2.7 or 2.8-times greater in cows with clinical mastitis during early gestation compared to cows without clinical mastitis in two investigations [24]. Exposure to clinical mastitis at any stage during lactation was linked to an elevated incidence of PL in dairy cows in two other trials [23]. In two further investigations, the odds of PL were more than 3.5-times greater in cows with subclinical mastitis before gestation [24], or 1.2-times higher in cows with subclinical mastitis during early gestation [21] than in cows without mastitis. According to Risco et al. [21], 

Cows with clinical mastitis during the first 45 days of gestation had a 2.7-times higher risk of PL within the next 90 days after mastitis diagnosis than cows without mastitis. Subclinical mastitis can cause a systemic inflammatory response, which can interfere with follicular expansion, oocyte quality, and embryo survival [23]. The odds of PL were 1.2-times higher in cows with subclinical mastitis during the first 90 days of gestation compared to cows without subclinical mastitis, according to Pinedo et al. [25].

We found an association between mastitis caused by *Escherichia coli*, *Staphylococcus aureus* and pathogenic Staphylococci and stillbirth. Further, the greatest difference between identified mastitis agents in cows by parity was found in cows with *Staphylococcus aureus, Escherichia coli*, and serogroup C Streptococci pathogens, in which the stillbirth of calves at the first calving was 1.38–1.65-times higher than cows of parity ≥ 2. Parity significantly reduced the risk of stillbirth in cows with *Staphylococcus aureus* and *Streptococcus agalactiae* mastitis at the end of pregnancy (0.630–0.716-times). According to the literature, bacterial proliferation, the release of endotoxins and exotoxins, and the release of inflammatory mediators are all implicated in the production of inflammatory mediators, which could contribute to luteolysis [26]. Prostaglandins, histamine, leukotrienes, and serotonin have been demonstrated to enhance cases of experimentally induced mastitis via intravenous infusions of lipopolysaccharides endotoxins (LPS) or intramammary infusions of *Escherichia coli* endotoxin [27]. Furthermore, investigations have demonstrated that Gram-negative bacteria can synthesize luteolytic prostaglandins after an infusion of endotoxins or septicaemia [21].

According to the literature, mastitis induced by Gram-negative bacteria can result in bacteraemia in more than 30% of affected cows [28]. Gram-positive bacteria have a cell wall made of multiple layers of peptidoglycan mucopeptide; these do not have endotoxin, but their presence in the mammary gland causes an inflammatory reaction very similar to that generated by Gram-negative bacteria’s endotoxins [29]. As a result, the data demonstrating the impact of mastitis on the rates of conception, early embryonic death, and abortions are evident [21]. Mastitis has been reported to alter the pattern of hormonal secretion and follicular development by releasing chemicals that inhibit the expression of receptors for gonadotropins and other reproduction-related hormones [30]. Other reproductive abnormalities, such as diminished oestrus expression and irregular cyclicity, may arise in situations with Gram-negative bacterial mastitis [31].

The study results showed that cows with subclinical mastitis caused by mixed microbiota also had stillbirths of calves. Subclinical mastitis caused by two or more different agents was previously described in Poland and Ireland [32], but to the best of our knowledge, there is no information about the impact on stillbirths of calves. Infection with mixed microbiota might have different effects—it can help both pathogens to escape initial immune surveillance and increase both pathogens’ transmission to the host or can cause decreasing overall virulence in pathogens [33]. More studies should be undertaken to establish the influence of mixed infection on stillbirths.

A limitation of our study was that most of the available research reported risk factors during the dry period for common transition diseases, such as subclinical mastitis, but there is limited information regarding how cows with mastitis during the last gestation period may be unfavourably associated with stillborn calves. Our study was conducted in herds with the same level of productivity in two categories, depending on the size of the herd. Factors, such as milk yield, herd size and other factors, which have an impact on subclinical mastitis, detected during the last gestation, and stillbirth in dairy calves should be evaluated in a larger study with more herds. One limitation of this study is the point prevalence of subclinical mastitis. As subclinical mastitis may differ over lactation, point prevalence may not have detected all cows with subclinical mastitis. Although unlikely to have a significant effect, this may have biased our results. 

Therefore, according to our knowledge, this is the first study that evaluates the relation of subclinical mastitis detected during the last gestation period (from the 210th day of pregnancy) with stillborn of calves. Therefore, our results suggest that the decreasing incidence of subclinical mastitis during the last gestation period (from the 210th day of pregnancy) can decrease the risk of stillbirth in dairy calves. The highest risk of stillbirth was found in cows with mastitis caused by *Escherichia coli*, *Staphylococcus aureus, Streptococcus agalactiae*, pathogenic Staphylococci, and other Streptococci. Cows at the first calving had a 1.38–1.65-times higher risk of stillbirth than in cows of parity ≥ 2.

## 5. Conclusions

This study provides new evidence that subclinical mastitis during the last gestation period affects the risk of stillbirth. Overall, this study shows that the late gestation period is challenging for stillbirth in the next lactation. Collectively, these results suggest that decreasing the incidence of subclinical mastitis during the last gestation period (from the 210th day of pregnancy) can decrease the risk of stillbirth in dairy calves. Further, it is important to identify the pathogen because the highest risk of stillbirth was found in cows with mastitis caused by *Escherichia coli*, *Staphylococcus aureus, Streptococcus agalactiae*, pathogenic Staphylococci, and other Streptococci. Cows at the first calving had a 1.38–1.65-times higher risk of stillbirth than in cows of parity ≥ 2. From a practical perspective, veterinarians and farmers can consider the effect of subclinical mastitis during late gestation on the risk of stillbirth and this could help with strategies for optimizing reproductive performance in dairy cows. Further studies should focus more on the impact of milk yield, herd size, and other factors on subclinical mastitis detected during the last gestation and stillbirth in dairy calves.

## Figures and Tables

**Figure 1 animals-12-01394-f001:**
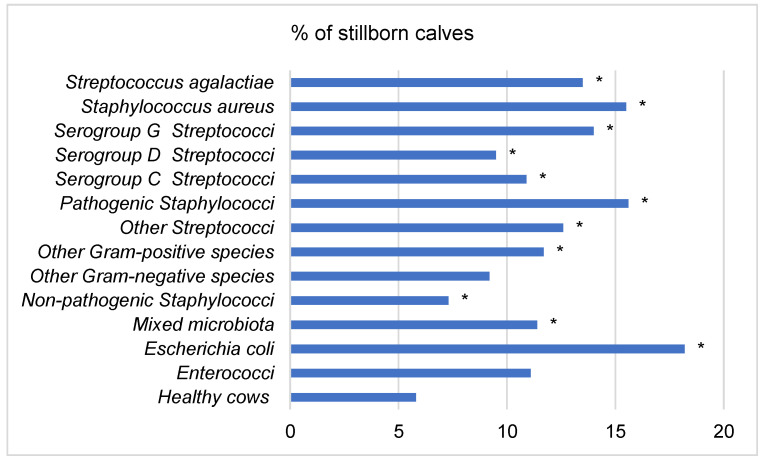
Percentage of stillborn calves in groups of cows. *—The difference within the group of healthy cows is statistically significant (*p* < 0.05).

**Figure 2 animals-12-01394-f002:**
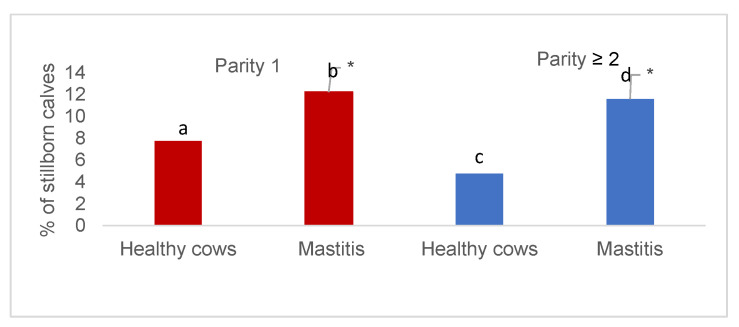
Percentage of stillborn calves by parity and groups of cows by health condition. *—the difference between the frequencies is statistically significant (*p* < 0.05).

**Figure 3 animals-12-01394-f003:**
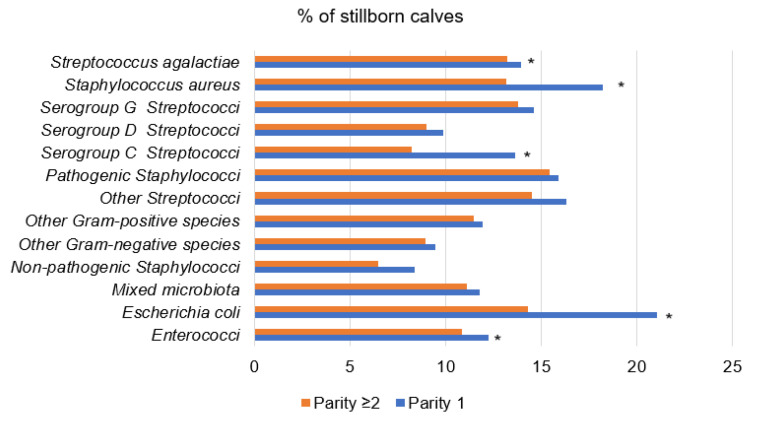
Percentage of stillborn calves by group of cows according to mastitis agent and parity. *—the difference between the frequencies of the parities groups is statistically significant (*p* < 0.05).

**Table 1 animals-12-01394-t001:** Description of herd characteristics.

Herd	Herd Size	Average Milk Production (Kg/Year)	AMS Units	Number of Sick Cows	Number of Healthy Cows
**Herd 1**	105	8500	Lely Astronaut^®^ A3 robots with free traffic	242	322
**Herd 2**	111	8300	Lely Astronaut^®^ A3 robots with free traffic	234	341
**Herd 3**	154	9500	Lely Astronaut^®^ A3 robots with free traffic	324	387
**Herd 4**	134	8750	Lely Astronaut^®^ A3 robots with free traffic	331	396
**Herd 5**	128	10,300	Lely Astronaut^®^ A3 robots with free traffic	344	499
**Herd 6**	201	12,000	Lely Astronaut^®^ A3 robots with free traffic	317	367
**Herd 7**	354	9800	Lely Astronaut^®^ A3 robots with free traffic	389	586
**Herd 8**	289	11,200	Lely Astronaut^®^ A3 robots with free traffic	398	597
**Herd 9**	365	9600	Lely Astronaut^®^ A3 robots with free traffic	483	674
**Herd 10**	545	11,350	Lely Astronaut^®^ A3 robots with free traffic	520	701

**Table 2 animals-12-01394-t002:** Association of mastitis with stillborn calves and parity of dairy cows.

Risk Factor	B	S.E.	Wald	df	*p*	OR	95% C.I. for OR	
							Lower	Upper
Group of cows	0.383	0.103	13.783	1	<0.001	1.467	1.198	1.796
Constant	−3.168	0.230	188.952	1	<0.001	0.042		

Groups of cows: 0—healthy cows, 1—cows with the identified mastitis; B—unstandardized regression weight; S.E. B—standard error for B, Wald χ^2^—this is the statistical test for the individual predictor variable; df—degrees of freedom, *p*—*p*-value (statistically significant with a *p*-value < 0.05); OR—odds ratio.

**Table 3 animals-12-01394-t003:** Odds ratio for association of stillborn calves with mastitis pathogens and parity of dairy cows.

Causative Agents of Mastitis	Factor	P	OR	95% C.I.for OR	
*Escherichia coli*	Group of cows	0.006	2.667	1.318	5.397
Mixed microbiota	Group of cows	<0.001	1.630	1.351	1.965
Non-pathogenic Staphylococci	Parity	0.050	0.658	0.433	1.001
Other Gram-negative species	Group of cows	0.036	1.264	1.015	1.573
Other Gram-positive species	Group of cows	<0.001	1.661	1.316	2.096
Other Streptococci	Group of cows	<0.001	2.504	1.566	4.004
Pathogenic Staphylococci	Group of cows	<0.001	2.991	2.027	4.415
*Staphylococcus aureus*	Group of cows	<0.001	2.905	2.034	4.149
	Parity Herd class	0.049 0.050	0.716 1.278	0.509 1.063	1.008 1.521
*Streptococcus agalactiae*	Group of cows	<0.001	2.469	1.509	4.037
	Parity	0.048	0.630	0.395	1.004

Groups of cows: 0—healthy cows, 1—cows with the identified mastitis agent indicated in the relevant row of the table. Two categories of parities (parity 1 and parity ≥ 2); herd class (100–200 cows and 201–600 cows); *p*-value (statistically significant with a *p*-value < 0.05); OR—odds ratio; 95% C.I. for OR—95% confidence level for odds ratio.

## Data Availability

The data presented in this study are available within the article.

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
