# Peer review of "Subclinical Mastitis Detected during the Last Gestation Period Can Increase the Risk of Stillbirth in Dairy Calves"

_animals, 2022, doi:10.3390/ani12111394_

Round 1

Reviewer 1 Report

Please find my considerations in the article pdf attached.

No conclusions were drawn based on the bibliography, only results.

Repeated phrases with very similar meanings

Materials and methods very poor in relation to the health status of animals and farms

Discussion must be improved

Author Response

Dear Reviewer,

Authors are very thankful with the comments, which help us to improve the manuscript. All changes proposed have been included in the manuscript and highlighted in yellow and track changes.

Best Regards,

Prof. Ramunas Antanaitis.

Question

Answers

Please find my considerations in the article pdf attached.

No conclusions were drawn based on the bibliography, only results.

Repeated phrases with very similar meanings

Materials and methods very poor in relation to the health status of animals and farms

Discussion must be improved

We corrected conclusion section – “This study provides new evidence that subclinical mastitis during the last gestation period affects the risk of stillbirth. Overall, this study shows that the late gestation period is challenging for stillbirth in next lactation.  Collectively, these results suggest that decreasing incidence of subclinical mastitis during the last gestation period (from the 210th day of pregnancy) can decrease the risk of stillbirth in dairy calves. Also, it is important to identified the pathogen because the highest risk of stillbirth was found in cows with mastitis caused by Escherichia coli, Staphylococcus aureus and pathogenicStaphylococci. Cows at the first calving was 1.38-1.65 times higher risk of the stillbirth of calves than in cows of parity ≥ 2. From practical point, veterinarians and farmers can consider the effect of subclinical mastitis during late gestation on  the risk of stillbirth and it it could help for strategies of optimizing reproductive performance in dairy cows. Further studies should focus more on the impact of milk yield, herd size and others factors on subclinical mastitis detected during the last gestation and stillbirth in dairy calves"

Also, we corrected introduction, MM, results sections, corrected and added new information in discussion section and we rewrote conclusion section.

from 210 till 260, define end of gestation for S. aures and Strep agalactiae

We corrected abstract section and added information – “Our results suggest that decreasing incidence of subclinical mastitis during the last gestation period (from the 210th day of pregnancy) can decrease the risk of stillbirth in dairy calves. Also, it is important to identified the pathogen because the highest risk of stillbirth was found in cows with mastitis caused by Escherichia coli, Staphylococcus aureus and pathogenicStaphylococci. Cows at the first calving was 1.38-1.65 times higher risk of the stillbirth of calves than in cows of parity ≥ 2. From practical point, veterinarians and farmers can consider the effect of subclinical mastitis during late gestation on the risk of stillbirth and it it could help for strategies of optimizing reproductive performance in dairy cows”

breastfeeding in woman

We corrected to – „When the negative impacts of stillbirth on the cows milking performance are included…”

higher, greater…

We corrected to  - „..significantly higher”

review the meaning of the sentence “ lower risk of fertilization”

- lower the risk of ovulatory infertility or of the oocyte being fertilized

We corrected to – „...lower the risk of ovulatory infertility or of the oocyte being fertilized”

lower survival is not stillbirth

increased days open???

We deleted this sentence

I don't understand the relevance of this paragraph to the mastitis theme

We deleted this sentence

something is missing

We corrected to – “…mastitis and stillbirth…”

1st time abbreviation in the text not explained PL; AI

We corrected to – “45 to 270 d after artificial insemination (AI))…”

And

“..mastitis on pregnancy loss (PL)..”

rewrite the 2 sentences

We corrected to – “According to this, we hypothesized that subclinical mastitis detected during the last gestation period can increase the risk of stillbirth and the aim was to investigate the relation of subclinical mastitis detected during the last gestation period and with stillborn of calves”

no reference to the method utilized

We corrected and added reference - Oliver, S. P., Gonzalez, R. N., Hogan, J. S., Jayarao, B. M., & Owens, W. E. (2004). Microbiological procedures for the diagnosis of bovine udder infection and determination of milk quality. Verona: National Mastitis Council, 47.

according to? no reference

We corrected and added reference - Kroger, D., & Jasper, D. E. (1967). Effect of milk age, storage, and testing temperatures upon the Wisconsin Mastitis Test score. Journal of Dairy Science50(6), 833-836.

and the SCC measuremen

We corrected and added reference - Krukowski, H., Lassa, H. E. N. R. Y. K. A., Zastempowska, E., Smulski, S. E. B. A. S. T. I. A. N., & Bis-Wencel, H. (2020). Etiological agents of bovine mastitis in Poland. Med Weter76(04), 221-5.

it is important to explain wich diseases or othar variables can influence the dystocia or the death in the peripartum

We added information – “Mee et al. (2014) found that perinatal mortality in dairy cattle is increasingly being recognized as a welfare issue. The most important modifiable variables impacting the risk of perinatal mortality are those that increase the likelihood of dystocia. Modifiable predictors are mostly (age at first calving, breeding method, sire, calving management, fetomaternal health status, and gestational nutrition) or moderately (calf breed, sex, gestation length) under the control of the dairy farm manager and thus can be changed to reduce the severity of this problem (Mee et al., 2014).

use the same words

We corrected to – “We found that the number of stillborn calves from healthy cows (without subclinical mastitis detected during the last gestation period) was 1.58 times lower in primiparous and 2.45 times lower in multiparous cows compared to the mastitis group by appropriate parity”

we are not informed about concomitant diseases or clinical signs like fever in the animals used

we dont know if the cows were vaccinated, or had brucelosis that cause abortion etc

We deleted this sentence – “It has been concluded that the usage of bacterial endotoxins results in dose-dependent clinical symptoms ranging from brief fevers to abortions 24 to 48 hours after intravenous administration”

they dont give a plasuble explanition for waht they found, or try to justify with bibliograph

We generally corrected discussion and conclusion sections.

Reviewer 2 Report

The authors address an interesting and important subject, and their results imply interesting associations between different mastitis pathogens and stillbirth in dairy calves. 

However, the manuscript needs substantial improvements before it is suitable for publication. My main issue with the manuscript is the Materials and Methods section and the statistical methods used. The inclusion of cows is not described clearly, and it does not seem like the clustering effect of herd is considered in the statistical analysis.

Major comments:

Cows within a herd will always be more alike than cows from different herds, which should be considered in all observational studies in dairy herds (i.e. your observations are not independent). For this particular study this is even more important as both infection pressure and environmental factors differ between herds and may effect both the prevalence of subclinical mastitis and the occurence of stillborn calves. You need to include herd as a random or fixed variable in your statistical analysis. I also think you should describe the included 10 herds in terms of herd size and production system, and provide information on how many cows you included from each herd.

It could also be argued that factors such as herd size and milk yield could be confounding effects and thus appropriate to include in the statistical models.

How did you evaluate model fit after the statistical analyses?

The grouping of cows is unclear to me. How many cows were screened with CMT? You state that they were at least 210 days pregnant, but how close to dry-off could they be tested? This may have an enormous effect on the CMT results as SCC rises as milk production declines.

It is stated that CMT test was used to group cows into "subclinical mastitis" and "no confirmed mastitis", but you don't define what a positive CMT test means. Is that also according to NMC? I think you at least should provide a link to a current web page where CMT result definitions can be found (I could not find them on the NMC official web page) or a reference, and also state how the distinction was made. I believe that a CMT test result of 3 (and possibly 2?) is indicative of subclinical mastitis? How were cows with a test result of more than 3, and cows with signs of clinical mastitis handled?

Were any cows treated with antibiotics or other treatments at dry-off (or before)?

You cultured milk from CMT positive cows, and investigated of cell count based on flow cytometry. You also checked the cell count from negative CMT tests with flow cytometry? After the culture results you state that groups of cows were "finally" formed, with cows with "isolated pathogens" assigned to the group of subclinical mastitis. It is also stated that these cows had a SCC exceeding 200 000. This process needs to be described more clearly, what happened to cows with a positive CMT result but no isolated pathogen? Were they excluded? Or cows with non-consistent results (i.e. positive CMT result but flow cytometry result of less than 200 000 or the other way around (negative CMT test but a cell count above 100 000)? 

I suggest that you make a flow chart or similar to describe how many cows you screened, and how many that were in the final analyses (and the reasons for cows being excluded in between). 

565 cows were diagnosed with "mixed microbiota". Is an infection with a mixed microbiota plausible, or could these findings be related to contamination at milk sampling? 

I agree that parity is a very important factor to consider in the analyses. However, if your aim is "to investigate the relation of subclinical mastitis [...] and its pathogens with stillborn calves", it might be better to stratify your dataset by parity, as it is well known that stillbirth is more common in primiparous cows. 

Were all cows of the same breed? Which?

The discussion needs rewriting. At several sections it is not clear if you are discussing your own results or others. The results of this particular study should be discussed in relation to previous research, with potential explanations for differences or similarities. Potential bias and weaknesses of the study should also be addressed. 

Minor comments: 

Introduction: Make sure you explain all abbreviations (see line 74), and use them instead of the full word once they are explained (see line 67 for example). It would also be nice if you defined subclinical and clinical mastitis and the distinction between those and intramammary infection (see NMC guidelines).

L 57-58 Seems like you should add "Jordanian" somewhere in that sentence.

L 62 "between subclinical mastitis and" what? The following sentences are very unclear and difficult to follow, you need to explain the study you are referring to in a better way.

L 67-70 Clarify if this was an established causal relationship and the chronology of events.

M&M:

L 107: "gestaton period of at least 260 days?

L  132-134: Not relevant here? 

Results:

Figure 1: The use of letters is confusing, it would be better to indicate significant results in some other way. Which statistical analyses are these P-values based on? (That information should be available in the Figure legend, same for all Figures and Tables). 

Author Response

Dear Reviewer,

Authors are very thankful with the comments, which help us to improve the manuscript. All changes proposed have been included in the manuscript and highlighted in yellow and track changes.

Best Regards,

Prof. Ramunas Antanaitis

Question

Answers

The authors address an interesting and important subject, and their results imply interesting associations between different mastitis pathogens and stillbirth in dairy calves. 

However, the manuscript needs substantial improvements before it is suitable for publication. My main issue with the manuscript is the Materials and Methods section and the statistical methods used. The inclusion of cows is not described clearly, and it does not seem like the clustering effect of herd is considered in the statistical analysis.

We have adjusted and supplemented the methodology section and analysis in the light of the comments provided

Major comments:

Cows within a herd will always be more alike than cows from different herds, which should be considered in all observational studies in dairy herds (i.e. your observations are not independent). For this particular study this is even more important as both infection pressure and environmental factors differ between herds and may effect both the prevalence of subclinical mastitis and the occurence of stillborn calves. You need to include herd as a random or fixed variable in your statistical analysis. I also think you should describe the included 10 herds in terms of herd size and production system, and provide information on how many cows you included from each herd.

In selected herds, the average somatic cell count in milk over the past three months has been above 200,000 cells/ml, and the average cow herd production has exceeded 8,000 kg of milk. The dry period of cows lasted from 45 to 60 days.

The cows in the herds were milked by Lely Astronaut® A3 robots with free traffic. The cows were of Holstein breed and their feed ration on all farms was balanced to match the energy and nutrient requirements of a Lithuanian Black-and-White cow weighing 550-650 kg, producing an average of 30 kg of milk per day.

The herds of cows were divided into two classes according to the average annual number of the herd (the first class - 100-200 cows (the first-fifth herds), the second - 200-600 cows (the sixth-tenth herds)).

Healthy cows in terms of reproductive status were similar to those with mastitis.

We evaluated cows in one herd during the study period: 322 to 701 with established mastitis and 234 to 520 healthy. Samples of sick and healthy animals in the herds were: 242 and 322 (Herd 1), 234 and 341 (Herd 2), 324 and 387 (Herd 3), 331 and 396 (Herd 4), 344 and 499 (Herd 5), 317 and 367 (Herd 6), 389 and 586 (Herd 7), 398 and 597 (Herd 8), 483 and 674 (Herd 9), 520 and 701 (Herd 10) cows, respectively.

All farms used a zero grazing system.

It could also be argued that factors such as herd size and milk yield could be confounding effects and thus appropriate to include in the statistical models.

Herd productivity was similar. We additionally used the herd size factor in the statistical analysis. The data are presented in the revised table 2.

How did you evaluate model fit after the statistical analyses?

We adjusted the goals of the work and the models accordingly.

The aim was to investigate the relation of subclinical mastitis detected during the last gestation period and its pathogens with stillborn of calves, considering that parity and herd size may also affect this result.

The grouping of cows is unclear to me. How many cows were screened with CMT? You state that they were at least 210 days pregnant, but how close to dry-off could they be tested? This may have an enormous effect on the CMT results as SCC rises as milk production declines.

Healthy and diseased cows were analogous in terms of reproductive status. All cows in the experiment were examined by the CMT method.

It is stated that CMT test was used to group cows into "subclinical mastitis" and "no confirmed mastitis", but you don't define what a positive CMT test means. Is that also according to NMC? I think you at least should provide a link to a current web page where CMT result definitions can be found (I could not find them on the NMC official web page) or a reference, and also state how the distinction was made. I believe that a CMT test result of 3 (and possibly 2?) is indicative of subclinical mastitis? How were cows with a test result of more than 3, and cows with signs of clinical mastitis handled?

We corrected  - “For this study, we used the most commonly used SCC diagnostic, the California Mastitis Test and various CMT score cutoff points were utilized to determine a positive CMT reaction (Dingwell et al. 2003). The single milk bacteriological culture result was used as the gold standard for calculating diagnostic test features (Dingwell et al. 2003). CMT were performed on each udder quarter of all cows. The CMT results were classified as negative (0+) or positive (1+) (traces), 2+ (gel), and 3+. (clumps) (Busato et al. 2000). Milk samples were collected aseptically from CMT >1+ quarters and submitted for somatic cell counting (SCC), bacteriological testing…”

Were any cows treated with antibiotics or other treatments at dry-off (or before)?

We corrected – “Cows with a positive CMT reaction were treated to all quarters with Rilexine DC (375 mg of cephaleksin, Virbac S.A.1eÌ€re Avenue, 2065 m, L.I.D. 06516 Carros, France). Internal teat sealant (ITS) containing bismuth subnitrate was infused into all quarters of all cows(Orbeseal, Zoetis). Following the final milking, antibiotic and ITS infusions were given as follows: teat ends were cleansed for at least 5 seconds with 70% isopropyl alcohol–soaked cotton swabs by trained staff wearing clean disposable gloves before the antibiotic treatment was infused into the mammary gland and again before ITS was infused into the teat cistern”

You cultured milk from CMT positive cows, and investigated of cell count based on flow cytometry. You also checked the cell count from negative CMT tests with flow cytometry? After the culture results you state that groups of cows were "finally" formed, with cows with "isolated pathogens" assigned to the group of subclinical mastitis. It is also stated that these cows had a SCC exceeding 200 000. This process needs to be described more clearly, what happened to cows with a positive CMT result but no isolated pathogen? Were they excluded? Or cows with non-consistent results (i.e. positive CMT result but flow cytometry result of less than 200 000 or the other way around (negative CMT test but a cell count above 100 000)? 

Cows with a positive CMT result but no isolated pathogen were excluded from this study. We found a total of 14 such cows.

I suggest that you make a flow chart or similar to describe how many cows you screened, and how many that were in the final analyses (and the reasons for cows being excluded in between). 

Healthy and diseased cows were selected based on reproductive status. We described the sample size of cows in the text.

565 cows were diagnosed with "mixed microbiota". Is an infection with a mixed microbiota plausible, or could these findings be related to contamination at milk sampling? 

We added information – “Bacteria isolated from Drigalski and Chromocult Coliform agar, as well as former dark blue or purple colonies, were identified as Escherichia coli. Mixed microbiota refers to samples that contain more than one microorganism species. (NMC (1999) and Krukowski et al., 2020)”

I agree that parity is a very important factor to consider in the analyses. However, if your aim is "to investigate the relation of subclinical mastitis [...] and its pathogens with stillborn calves", it might be better to stratify your dataset by parity, as it is well known that stillbirth is more common in primiparous cows. 

We tried to analyze the influence of factors in one model so that the volume of our article does not increase dramatically. We would like to take advantage of your offer in the future as we experiment with a larger homogeneous sample.

Were all cows of the same breed? Which?

We added information – “Throughout the trial period, the feed rations on all farms were balanced to meet the energy and nutrient requirements of a 550–650 kg Lithuanian Black-and-White cow producing 30 kg/day of milk on average

The discussion needs rewriting. At several sections it is not clear if you are discussing your own results or others.

The results of this particular study should be discussed in relation to previous research, with potential explanations for differences or similarities.

Potential bias and weaknesses of the study should also be addressed. 

We corrected discussion and conclusion sections L249 - 420

Minor comments: 

Introduction: Make sure you explain all abbreviations (see line 74), and use them instead of the full word once they are explained (see line 67 for example). It would also be nice if you defined subclinical and clinical mastitis and the distinction between those and intramammary infection (see NMC guidelines).

We corrected - pregnancy loss (PL)

We added information – “Clinical mastitis is defined as "abnormal milk," hence no mention of SCC is necessary. (Kitchen et al). However, clinical mammary tissue will nearly invariably contain SCC with more than 200,000 cells per milliliter. Milk from healthy, uninfected mammary glands is white to whitish-yellow in color and is devoid of flakes, clots, and other obtrusive modifications in appearance. The great majority of these defects are caused by mammary gland bacterial infection. In general, the aberrant appearance of the secretion from the infected area increases with the severity of the infection. When quarterly SCC are equivalent to or greater than 200,000 cells/ml and bacteria are identified in the absence of clinical changes, the quarter is considered subclinical (NMC guidelines)

L 57-58 Seems like you should add "Jordanian" somewhere in that sentence.

We corrected

L 62 "between subclinical mastitis and" what? The following sentences are very unclear and difficult to follow, you need to explain the study you are referring to in a better way.

We corrected to – “…between subclinical mastitis and stillbirth was…”

L 67-70 Clarify if this was an established causal relationship and the chronology of events.

We added information - … Dystocia impaired lactation performance…”

And – “The immune response to a mammary gland infection is of highest relevance for the dairy cow's health. Harmon (1994) observed that the rise in milk SCC is due to the migration of polymorphonuclear cells from the blood vessels to the mammary gland as a result of the production of inflammatory mediators. This mechanism may be comparable in animals with subclinical mastitis, resulting in diminished reproductive performance (Schrick et al. 2001)”

M&M:

L 107: "gestaton period of at least 260 days?

We corrected to – “..gestation period at least 260 days..”

L  132-134: Not relevant here? 

We moved this sentence – “Throughout the trial period, the feed rations on all farms were balanced to meet the energy and nutrient requirements of a 550–650 kg Lithuanian Black-and-White cow producing 30 kg/day of milk on average” to Location and animals part.

Results:

Figure 1: The use of letters is confusing, it would be better to indicate significant results in some other way. Which statistical analyses are these P-values based on? (That information should be available in the Figure legend, same for all Figures and Tables). 

We have adjusted in line with the comments made.

Round 2

Reviewer 1 Report

Line 33 - identify

line 37 - 2x it

line 48 - "lower risk of"

line 51 - rewrite to make sense

line 53 - mortality

line 58 - clinical mastitis = clinical signs...don´t understand why "hence no mention of SCC is necessary", but in the next sentence again SCC. Why talk about the milk, if what matters is the general clinical signs that may lead to SB? 

line 86 - here well explained de inflammatory response and reason to the rise in SCC during IMI, could be in the other paragraph...

line - 130 - somatic cell count/number is repeated

line 159 - reference?

line 212 - with or within?

line 268 - there is some information missing or rewrite

280 - Attention!!! copy past from the following article DOI:10.1016/j.tvjl.2013.08.004

Author Response

Dear Reviewer,

Authors are very thankful with the comments, which help us to improve the manuscript. All changes proposed have been included in the manuscript and highlighted in yellow and track changes.

Best Regards,

Prof. Ramunas Antanaitis.

Question

Answers

Line 33 - identify

We corrected to – “….important to identify the pathogen..

line 37 - 2x it

Deleted second “it”

line 51 - rewrite to make sense

We corrected to – “The negative effect of stillbirth on milk production was greatest early lactation [5]”

line 53 - mortality

We corrected to – “..available for sale, calf mortality is an important…”

line 58 - clinical mastitis = clinical signs...don´t understand why "hence no mention of SCC is necessary", but in the next sentence again SCC. Why talk about the milk, if what matters is the general clinical signs that may lead to SB? 

We deleted this sentence – “Clinical mastitis is defined as "abnormal milk," hence no mention of SCC is necessary. [9]”

line 86 - here well explained de inflammatory response and reason to the rise in SCC during IMI, could be in the other paragraph...

We moved this sentence to other paragraph.

line - 130 - somatic cell count/number is repeated

We corrected – “…submitted for somatic cell counting (SCC), bacteriological testing in milk (using Somascope, CA-3A4, Delta Instruments”

ine 159 - reference?

We corrected and added reference – “Internal teat sealant (ITS) containing bismuth subnitrate was infused into all quarters of all cows (Orbeseal, Zoetis). Following the final milking, antibiotic and ITS infusions were given as follows: teat ends were cleansed for at least 5 seconds with 70% isopropyl alcohol–soaked cotton swabs by trained staff wearing clean disposable gloves before the antibiotic treatment was infused into the mammary gland and again before ITS was infused into the teat cistern [10].

References – 10. National Mastitis Council. Laboratory handbook on bovine mastitis. Rev. ed. Madison, WI : National Mastitis Council, 1999.

line 212 - with or within?

We corrected to. – “- The difference within the group of healthy…”

line 268 - there is some information missing or rewrite

We corrected to – “. According to our knowledge the most of the available research reporting the impact of risk factors during the dry period on diseases after calving such as subclinical mastitis, but…”

280 - Attention!!! copy past from the following article DOI:10.1016/j.tvjl.2013.08.004

We corrected to – “By controlling predictors such as breeding method, age at first calving, calving management and ect.), we can reduce the severity of this problem [24]”

Reviewer 2 Report

Thank you for addressing some of my concerns. However, grouping the herds into two groups depending on herd size is not sufficient to account for differences between different herds, see my previous comments regarding this.  Again, I think you should include herd as a random variable in your statistical models (or motivate why you don't). Also, the grouping is very rough (especially when it´s not clear how the herds vary in size), herds with 201 cows would be more similar to herds with 199 cows compared to herds with 600 cows. Did you try to use herd size as a continuous variable in the analyses instead? 

Did you evaluate model fit by looking at residual plots for the logistic regression model?

The section on herd characteristics added in the M&M section (line 108-114) is rather inclusion criteria than a description of the herds. I understand that you chose the herds to be similar to one another which is good, but I would like to know how they vary. Exceeding 8,000 kg milk per cow and year could mean that they produced between 8,000 and 9,000 within the dataset, or between 8,000 and 13,000. I would prefer a table describing herd characteristics including herd size, average milk production, AMS units and number of sick and healthy animals (also as line 142-146 is now very difficult to read). 

I understand that mixed microbiota refers to samples with growth of more than one species. My question is if these samples are cases of subclinical mastitis or if it is more likely a contamination of samples? That should at least be included in the discussion.

The language overall, and especially in the revised sections, has to be improved.

Author Response

Dear Reviewer,

Authors are very thankful with the comments, which help us to improve the manuscript. All changes proposed have been included in the manuscript and highlighted in yellow and track changes.

Best Regards,

Prof. Ramunas Antanaitis.

Question

Answers

Thank you for addressing some of my concerns.

However, grouping the herds into two groups depending on herd size is not sufficient to account for differences between different herds, see my previous comments regarding this. 

Again, I think you should include herd as a random variable in your statistical models (or motivate why you don't).

Also, the grouping is very rough (especially when it´s not clear how the herds vary in size), herds with 201 cows would be more similar to herds with 199 cows compared to herds with 600 cows.

Did you try to use herd size as a continuous variable in the analyses instead? 

Did you evaluate model fit by looking at residual plots for the logistic regression model?

The herds of cows were divided into two classes according to the size of the herd: class 1 - 100-200 cows (average 126 cows in the herd) class 2 - 201-600 cows (average  351  cows in the herd).

We corrected the statistical models of  logistic regression

Multivariable logistic regression models were used to analyze the factors influencing the likelihood of stillbirth in calves.

Estimates and Wald 95% limits were used to calculate odds ratios (OR) and 95% confidence intervals (CI). The Wald χ2 statistic was used to determine the significance of each variable in the model.

Using a backward stepwise logistic regression method we eliminated all non-essential explanatory variables. Variables were constantly removed from the models according to the significance of the Wald criterion. 

In this way, all final statistical models included only significant explanatory variables.

Herd size was used as categorical class variable.

The section on herd characteristics added in the M&M section (line 108-114) is rather inclusion criteria than a description of the herds. I understand that you chose the herds to be similar to one another which is good, but I would like to know how they vary. Exceeding 8,000 kg milk per cow and year could mean that they produced between 8,000 and 9,000 within the dataset, or between 8,000 and 13,000. I would prefer a table describing herd characteristics including herd size, average milk production, AMS units and number of sick and healthy animals (also as line 142-146 is now very difficult to read). 

Between 8000 and 12000

We added Table 1. Describtion of herd characteristics.

Herd

Herd size

Average milk production (kg/year)

AMS units

Number of sick cows

Number of healthy cows

Herd 1

105

8500

Lely Astronaut® A3 robots with free traffic

242

322

Herd 2

111

8300

Lely Astronaut® A3 robots with free traffic

234

341

Herd 3

154

9500

Lely Astronaut® A3 robots with free traffic

324

387

Herd 4

134

8750

Lely Astronaut® A3 robots with free traffic

331

396

Herd 5

128

10300

Lely Astronaut® A3 robots with free traffic

344

499

Herd 6

201

12000

Lely Astronaut® A3 robots with free traffic

317

367

Herd 7

354

9800

Lely Astronaut® A3 robots with free traffic

389

586

Herd 8

289

11200

Lely Astronaut® A3 robots with free traffic

398

597

Herd 9

365

9600

Lely Astronaut® A3 robots with free traffic

483

674

Herd 10

545

11350

Lely Astronaut® A3 robots with free traffic

520

701

I understand that mixed microbiota refers to samples with growth of more than one species. My question is if these samples are cases of subclinical mastitis or if it is more likely a contamination of samples? That should at least be included in the discussion.

Milk samples in which mixed microbiota was detected belong to cows with subclinical mastitis. Information was added to the discussion section under your advice.

We corrected - “Study result showed that cows with subclinical mastitis caused by mixed microbiota also had stillbirths of calves. Subclinical mastitis caused by two or more different agents was previously described in Poland [23] and Ireland [39], but at the best of our knowledge, there is no information about the impact on stillbirths of calves. Infection with mixed microbiota might have different effects -it can help both pathogens to escape initial immune surveillance and increase both pathogens’ transmission to the host or can cause decreasing overall virulence of pathogens [40]. More studies should be taken to find influence of mixed infection to stillbirths.“